# Significant Rise of Colorectal Cancer Incidence in Younger Adults and Strong Determinants: 30 Years Longitudinal Differences between under and over 50s

**DOI:** 10.3390/cancers14194799

**Published:** 2022-09-30

**Authors:** Dimitra Sifaki-Pistolla, Viktoria Poimenaki, Ilektra Fotopoulou, Emmanouil Saloustros, Dimitrios Mavroudis, Lampros Vamvakas, Christos Lionis

**Affiliations:** 1Department of Social Medicine, School of Medicine, Clinic of Social and Family Medicine, University of Crete, 700 13 Heraklion, Greece; 2Cancer Registry of Crete, School of Medicine, University of Crete, 700 13 Heraklion, Greece; 3Department of Oncology, University Hospital of Larissa, 413 34 Larisa, Greece; 4Department of Medical Oncology, University General Hospital of Heraklion, 715 00 Heraklion, Greece; 5Region of Crete, 712 01 Heraklion, Greece

**Keywords:** colorectal cancer, colon cancer, rectal cancer, risk factors, survival, metastasis, treatment, big-data, population-based cohort

## Abstract

**Simple Summary:**

We aimed to assess the incidence of CRC in Crete from 1992–2021 and compare them among younger and older adults. The mean age-specific incidence rate (ASpIR/100,000/year) of colon cancer patients <50 years was 7.2 (95% CI 5.1–9.7), while for patients ≥50 years the ASpIR was 149 (95% CI 146.2–153.4). ASpIR presented a 29.6% increase from 2001 to 2011 in the age group of 20–34 years and further increase is expected from 2022–2030 (projected change, 42.8%). The main risk factors were the pack years (*p* = 0.01), alcohol consumption (0.02), and farmer occupation (0.04), especially during 2012–2021. We confirmed an increased incidence of CRC in young adults <50 in a European population with low cancer incidence in the past and a worrisome prediction for the near future. There is a clear indication that starting CRC screening at an earlier age may be essential.

**Abstract:**

(1) Background: There is evidence in the recent literature that the incidence patterns of colorectal cancer (CRC) have changed considerably over the years, tending to rise rapidly in individuals under 50 years old compared with those over 50 years. The current study aimed to assess the incidence of CRC in Crete from 1992–2021 and compare them among younger and older adults. (2) Methods: Data on malignant neoplasms of colon, rectosigmoid junction, and rectum have been extracted from the database of the Regional Cancer Registry of Crete. (3) Results: The number of these cases for the period 1992–2021 was 3857 (*n* = 2895 colon and *n* = 962 rectum). The mean age-specific incidence rate (ASpIR/100,000/year) of colon cancer patients <50 years was 7.2 (95% CI 5.1–9.7), while for patients ≥50 years the ASpIR was 149 (95% CI 146.2–153.4). ASpIR presented a 29.6% increase from 2001 to 2011 in the age group of 20–34 years and further increase is expected from 2022–2030 (projected change, 42.8%). The main risk factors were the pack years (*p* = 0.01), alcohol consumption (0.02), and farmer occupation (0.04), especially during 2012–2021. (4) Conclusions: We confirmed an increased incidence of CRC in young adults <50 in a European population with low cancer incidence in the past and a worrisome prediction for the near future. The observed trends clearly indicate that starting CRC screening at an earlier age may be essential.

## 1. Introduction

Colorectal cancer (CRC) incidence and mortality has been presenting diverse trends among countries, with declining rates in many countries worldwide (e.g., the United States, New Zealand, northern European countries, etc.) for over a decade [1]. In the United States, CRC incidence rates had marked a decline of approximately 2 percent per year [2]. However, incidence in most western countries has remained stable or has increased slightly during the last decade [2]. Even more rapid significant increases have been monitored in many regions that were historically at low risk, including Spain, Italy, Greece, and various countries within eastern Asia and eastern Europe [3,4]. Still, in all cases, CRC remains the third most incident and fatal cancer globally, which indicates that it is still a major public health problem [5]. According to the World Health Organization GLOBOCAN database, CRC is the third most commonly diagnosed cancer in males and the second in females [5].

Focusing on this impressive decreasing pattern that has been monitored in many countries, this seems to be observed mainly in the older populations, whereas recent reports indicate a rising CRC incidence in younger adults (i.e., individuals younger than 50 years old) [2,6]. Furthermore, CRC is often diagnosed at a later stage in younger adults, when the disease is more challenging to treat, therefore these patients tend to have lower survival [6]. In a retrospective study of Mayo Clinic cancer registry (1972–2017), the investigators found an augmented tendency of CRC cases in younger adults (<50 years) and they noticed that more often it was located at the left-sided colon or at the rectum. One more finding that came out of their research was that a high proportion of the younger age group had an advanced rather than early-stage cancer diagnosis [7]. This is probably due to the delays in seeking medical care and misdiagnosis [6,7].

Additionally, there is some evidence supporting that this temporal rise in the onset of CRC incidence among individuals under 50 years may be attributable to obesity and other lifestyle risk factors [8]. Other contributors may be the genetic influences and changes in the environment and the relative exposures [9]. A very recent meta-analysis [10] revealed that significant risk factors for early-onset CRC included CRC history in a first-degree relative (relative risk (RR) 4.21), hyperlipidemia (RR 1.62), obesity (RR 1.54), and alcohol consumption (RR 1.71). Mechanisms contributing to increasing incidence rates are poorly understood and require further studies [6].

Overall, the latest literature insights are indicating an increase in the age-specific incidence rates, the age-weighted incidence rates, and the annual percentage change of the incidence in younger adults, stressing the emerging trends in this population group [6,7,8,9,10]. In the fight against CRC, screening is on the frontline since it is a strongly curable cancer when diagnosed early [10]. Strong and well-coordinated screening programs in countries with robust healthcare systems have already proved to be effective towards this direction, by stabilizing or decreasing CRC incidence and mortality rates [11]. For instance, the American Cancer Society signalizes [12] the hazard is lurking in younger adults and indicates that there is a reduction in CRC incidence in older individuals due to the high participation in screening tests and because they changed their lifestyle risk factors. In parallel, they stress the increased incidence among younger adults and the unknown causal factors. Consequently, they propose to start screening tests at a younger age (e.g., 45 years). In a retrospective study conducted in Michigan from 1998–2011, comparing age groups above and below the threshold for screening tests (50 years), 15% of the CRC patients were <50 years. These younger patients were more likely to develop advanced colorectal cancer, receive strong treatment, and achieve long-term survival. The authors highlighted the need to revise the screening protocol at younger ages and focus on enhancing lifestyle and cancer-related behaviors at a younger age [13].

In this regard, nationally representative databases are a useful tool to examine temporal incidence trends of CRC according to age and relation to several lifestyle and clinical risk factors [14]. Greece is a country where, despite the empirical observations, limited studies have evaluated the real CRC incidence burden and the screening coverage in the country. Some first indications were confirmed by the primary records of the Cancer Registry of Crete, Greece (www.crc.uoc.gr, accessed on 15 August 2022), which showed an increasing trend of the CRC incidence in the age group of <50 years during the last years. These observations and the existing evidence in the literature highlight the need for population-based studies on CRC trends among younger and older individuals in relation to core risk factors.

The current study aimed to assess the incidence of CRC in Crete during the last three decades (1992–2021) and compare them among younger and older adults (under and over 50 years old). Additionally, we explored the main established determinants for CRC incidence in young adults and tested whether they differed temporally. Lastly, we explored variations between younger and older adults with regards to stage at diagnosis, the type of treatment-intervention, and the course of the disease (i.e., multiple metastases, survival).

## 2. Materials and Methods

### 2.1. Study Design and Methodological Approach

This retrospective open population-based cohort study was conducted in Crete, Greece using the population-based Cancer Registry of Crete. The registry is a member of the International Association of Cancer Registries (IACR) and the European Network of Cancer Registries (ENCR) [15]. It aims to systematically and comprehensively record the cases and deaths from cancer on the island in order to propose valid methods, which concern the prevention as well as the management. It was founded in 1992 in Crete and includes all the counties of Crete. In recent years, the infrastructure and operation of the registry has been upgraded, and a new digital cancer monitoring and recording system (CMS, based on the CANReg 5) has been introduced. This is suitable for importing and managing big data in accordance with international disease coding standards (ICD10-O and ICD10 for chronic diseases), as well as for the protection of personal data.

### 2.2. Study Population and Sample Size

For the needs of this research, the study population was defined as the total number of permanent residents (total population based on the 2011 census: 623,065). Data on malignant neoplasms of colon (ICD-10: C18) and rectosigmoid junction and rectum (ICD-10: C19–20) were obtained from the database of the regional Cancer Registry of Crete for the period 1992–2021. Out of them, the total number of new cases that had a confirmed diagnosis of colon cancer (ICD-10: C18) and orthosigmoid ligament and rectum (ICD-10: C19–20) for the period 1992–2021 was 3857 (*n* = 2895 colon and *n* = 962 rectum).

### 2.3. Variables and Inclusion Criteria

Information on the patient’s demographic profile, personal and family medical history, and lifestyle factors (body mass index-BMI, body surface area-BSA, smoking habits, alcohol consumption) were also available.

Three main inclusion criteria were determined as follows: (a) cases with primary colon or rectal cancer, (b) individuals who have been residing in Crete for at least 10 years before the diagnosis, and (c) histologically confirmed diagnosis of colon or rectal cancer.

### 2.4. Data Quality Indicators

The Cancer Registry of Crete performs systematic quality controls for all collected data. Quality checks include all types of controls for duplicated or missing cases, technical or other errors, and issues of data accuracy, validity, and completeness according to the European guidelines [16]. Further details of these processes are provided in a previous publication of the registry [17,18,19]. In the current study of colon and rectal cancer, quality indices were very high: completeness = 98.5%, validity/accuracy = 97.5%, and timeliness = 99%.

### 2.5. Statistical Analysis

Initially, age-specific incidence rates per 100,000 residents per year (age-specific incidence rates-ASpIR/100,000/year) and age-weighted incidence rates per 100,000 inhabitants per year (Age-Standardized Incidence Rates-ASIR/100,000/year) were estimated through direct standardization, using the Cretan total population of 623,065 permanent residents and the standard European population [20]. These indicators were calculated for the total population in Crete by cancer type (colon and rectum cancer). The ASIR annual percentage change (APC) was estimated per year, while projection of the expected changes was attempted using a Bayesian model, as introduced by Riebler and Held [21].

Then, the distribution of all variables was checked through the Kolmogorov–Smirnov test and binomial test and indicated that non-parametric statistical tests were applied (*p*-value < 0.05). Therefore, we performed the Mann–Whitney and Kruskal–Wallis tests. Additionally, we developed a binary logistic regression model with adjustment to gender, place of residence, comorbidities of the gastroenterological system, and cancer family history, in order to estimate the determinants for CRC in younger adults versus older adults. This model was tested for all three different decades and the variations among them were assessed by the Mantel–Haenszel procedure (Yate’s corrected; Ho: p1 = p2). In addition to the aforementioned, net-survival (%) was considered through survival analysis and Kaplan–Meier Curves [22]. The analysis was performed in the Stata, while all tests were performed at a level of statistical significance α = 0.05 and were two-tailed.

### 2.6. Ethical Approval

The CRC holds a license from the Hellenic Data Protection Authority (Protocol number: 960/11-08-2009) and has adopted the rules for collecting, managing, and processing sensitive and personal data. All information was recorded using a cryptographic coding system in accordance with the federal law principles and stored in the CMS server. No personal or individual-level data are or will be published.

## 3. Results

A total of 3857 cases (*n*= 2895 colon and *n* = 962 rectal cases) with complete personal medical history were included in this analysis. The majority of the 3857 histologically confirmed patients were diagnosed with colon cancer (75.1%), while 24.9% were rectosigmoid junction and rectal cases.

Table 1 depicts the patient characteristics (N, %) and the APC (%, 95% CI). The colon ASIR increased significantly from 1992 to 2021 with a statistically significant annual percentage change of 1.5 (95% CI, 1.3–1.7). Similarly, rectal ASIR presented a lower annual percentage increase (APC, 0.9%; CI, 95%, 0.0–1.6). Males presented a higher incidence compared with females (55.5% and 44.5%, respectively); however, females had significantly higher increasing trends (APC, 1.8%; 95% CI, 1.4–2.2). Nevertheless, pronounced increases of the ASIR were observed in the younger age groups of 20–34 (APC, 1.8%; 95% CI, 1.2–2.6) and 35–59 years (APC, 1.6%; 95% CI, 1.3–1.9). In addition, significant decreases of the APC were observed in the older adults (50–74 years: APC, −1.1%; 95% CI, −1.4–−0.5%; >75 years: APC, −1.4% 95% CI, −1.3–−1.0). Stage at diagnosis and place of residence were also found to be associated with the APC of incidence rates (Stage II: APC, 2.2%; 95% CI, 1.9–2.2; country of Lasithi: APC, 1.9%; 95% CI, 1.7–2.3).

The mean ASpIR of colon cancer patients <50 years was 7.2/100,000/year (95% CI 5.1–9.7) (Figure 1A), while for patients ≥50 years the ASpIR was 149/100,000/year (95% CI 146.2–153.4) (Figure 1B). Males presented significantly higher rates compared with females (*p* = 0.01) especially between the age groups of 30–49 and > 70 years. On the contrary, females aged 20–24 years presented slightly higher ASpIR compared with males (Figure 1A). The ASpIR seem to start declining slightly after the age of 85–89 years old (Figure 1B).

Observed and projected annual change (%) of colon ASpIR per age group are presented in Figure 2. Statistically significant changes (%) were observed in most age groups (*p* < 0.05). In particular, ASpIR presented a 29.6% increase from 2001 to 2011 in the age group of 20–34 years and further increase is expected from 2022–2030 (projected change, 42.8%). Similar trends were observed in the age group of 35–49 years from 2001 to 2011. In contrast, the ages over 50 years presented declines in the ASpIR starting from −12.4% (2001–2011) to −25.8% (2012–2021) for individuals aged 50–74 years, and from −31.3% (2001–2011) to −39.6% (2012–2021) for those aged >75 years.

ASpIR of rectal cancer patients per age group presented almost similar trends with those of colon cancer patients. The mean ASpIR of rectal cancer patients <50 years was 4.2/100,000/year (95% CI 3.8–4.6) (Figure 3A), while for patients ≥50 years the ASpIR was 96.9/100,000/year (95% CI 95.1–98.0) (Figure 3A). Males presented significantly higher rates compared with females (*p* = 0.03), especially after the ages of 40, 75, and 85 years. The ASpIR presented decreasing trends after the age of 85 years (Figure 3B).

Furthermore, the observed and projected annual change (%) of rectal ASpIR per age group are presented in Figure 4. Similar to the colon cancer trends, several statistically significant changes (%) were observed after 2001 (*p* < 0.05). In the age group of 20–34 years, the ASpIR presented a 34.7%% increase from 2001 to 2011, with a projected increase of 52.1% from 2022 to 2030. In the age group of 35–49 years, similar trends were observed reaching a 37.2% increase from 2001 to 2011, with an estimated increase of more than 56% for the period 2022–2030.

Table 2 presents the core lifestyle risk factors for younger adults, accounting for gender, place of residence, comorbidities of the gastroenterological system, and cancer family history. The adjusted RRs are presented for the three different decades (1992–2000, 2001–2011, 2012–2021) and they are compared to identify interactions of the risk factors among the years. It seems that increased body mass index (BMI) and body surface area (BSA), smokers, increased pack years, alcohol consumption and number of glasses per week, as well as farmer occupation present increased probability for CRC incidence. Still, several variations were observed among the three time periods. More specifically, BMI presented a significantly increasing RR over the years, (1992–2000, 2001–2011, and 2012–2021: RR; 1.2, 1.8, 2.5, respectively, *p*-value = 0.04). BSA presented similar trends with a *p*-value for interaction < 0.001. Pack years (*p*-value for interaction = 0.01), alcohol consumption (0.02), and farmer occupation (0.04) presented significantly increasing trends since 2012–2021. Lastly, number of glasses of alcohol per week was also a statistically significant risk factor, with a RR that doubled from 1992 to 2021 (RR 1992–2000: 2.1 95% CI 1.9–2.3, RR 2012–2021: 4.1 95% CI 3.9–4.3).

Table 3 presents the statistically significant differences of several parameters between the two age groups over the years. It was found that the 5-year net-survival changed significantly between the two age groups among the three decades (*p*-value = 0.03). Particularly, younger adults tend to present lower survival as the years goes by, while older adults have rising survival trends. Additionally, younger adults presented higher percentages of diagnosis at a late stage (III and IV) from 2001 to 2021, contrary to the older adults (*p*-value = 0.03). Similarly, chemotherapy tends to be more frequent in younger adults during the last two decades compared to 1992–2000 (*p*-value = 0.04).

## 4. Discussion

### 4.1. Main Findings

The present population-based study assessed the temporal CRC trends in Crete, by utilizing data of three decades, and explored the main lifestyle risk factors among younger and older adults. Our core findings managed to enlighten the research questions we had set and are summarized as follows. (a) A significant rising trend of CRC in adults <50 years was observed for the period 1992–2021, especially after 2001. In parallel, CRC patients >50 years presented a decreasing incidence trend for the same period. More worrisome is the prediction that the incidence in individuals <50 years is expected to further increase in the years to come. (b) Additionally, the main lifestyle risk factors that were tested for young adults versus older adults and were found to significantly increase the probability of CRC incidence were the following: BMI and the BSA, smoking status, and alcohol consumption. In particular, individuals <50 years old with increased BMI and BSA, smokers with increased pack-years, heavy alcohol consumers, and farmers were found to present significantly higher odds for CRC during the last two decades. (c) Lastly, several other clinical parameters varied between <50 versus >50 years and presented significant temporal trends among the three decades that we studied. These parameters included 5-year net survival, which tended to be lower for young CRC patients after 2001 and 2012, late stage at diagnosis, and chemotherapy, which were more frequent in the <50 years after 2001.

### 4.2. Reflections of the Literature

Our results are in line with those of the international literature which have already noted the rising trends of CRC in younger adults during the past years, highlighting the changing scenery. According to Vuik et al., CRC incidence in young adults increased at 7.9% per year (or 79% per decade) in the specific age group (<50 years) [23]. Similar but slightly lower trends were seen for the under 50 Cretan population. The CRC incidence increased by approximately 3.3% per year (or 32.5% for the last decade). Overall, our temporal data are in accordance with published reports from other registries showing a similar trend, and with the key findings of a recent report showing that CRC incidence rates are uniquely increasing in young adults in nine high-income countries (Germany, USA, Australia, Canada, New Zealand, UK, Denmark, Slovenia, and Sweden) across North America, Europe, and Oceania, whereas rates in older adults are stable or declining [24]. They even reported that the largest increase in CRC incidence rate was observed in the age group of 20–39 years [23]. Furthermore, according to this study, CRC declined in young adults in Italy, while it increased in Cyprus; both are neighboring countries to Greece, sharing similar lifestyle patterns. Although these three countries share many similarities in terms of lifestyle and behavioral patterns, they still differ in CRC screening. Both Cyprus and Greece, which have increasing trends in CRC incidence rates especially in those under 50 years old, lack national CRC screening programs [25].

The causal mechanisms for this increasing incidence are still unknown, but the potential contributors may be related to the identified risk factors by our study and other evidence in the literature. These include obesity [8,9], smoking [26,27], alcohol consumption [28,29], occupational exposures [10,30], and others. Our findings on the rise of incidence in younger adults and the adverse contributors of BMI and BSA are obviously linked to the already-reported obesity epidemic in Greece [31], and lack of exercise and dietary habits [32,33], including consumption of more alcohol and processed meat [34]. These potential interactions of obesity and the changing patterns of lifestyle habits have also been reported for other countries [8]. The statistically significant changes of ASpIR (%) that were observed after 2001 in most age groups denotes the impact of the lifestyle changes of the Cretan population during the past few decades. To the best of our knowledge, population-based studies estimating incidence of CRC in Greece are lacking. Τhe only available study has reported constant increases of colorectal adenocarcinoma [35], as has been observed in many western countries [23].

From a clinical point of view, there are few studies sharing reflections on the biological and other clinical mechanisms. Still, we have observed differences in clinical measures that are in line with other studies. Focusing on metastatic CRC, we have observed increases in the proportion of young adults compared with older ones, but it is not statistically significant. Still, this is an interesting finding with a temporal changing pattern that may be even higher in the upcoming years. This is in line with another study [36] reporting that younger adults in comparison with older patients had greater proportions of synchronous metastatic disease (80.4% versus 64.4%, respectively; *p*-value = 0.04). Additionally, the percentage of patients under 50 years old receiving chemotherapy is rising and is already significantly higher than in older CRC patients. The study by Vatandoust et al., is again in compliance with our findings [36]. Stage at diagnosis may play a role in these observations. We monitored increased late stage at diagnosis especially among young adults. Other retrospective analyses have suggested that young CRC patients were also diagnosed at much later stages of the disease compared with older ones [37,38]. However, evidence on the prognosis of young patients with CRC varies in the literature. Some studies suggested that younger patients have worse prognoses [39,40,41], whereas others reported that the prognosis does not differ between younger and older adults [42,43], while for some stages of the disease such as stage IV, younger patients might have a better outcome due to more aggressive treatment [32,44,45].

Furthermore, Willauer et al. [46], recently assessed the clinical and molecular features of colorectal tumors in young adults compared with adults over 50 years old. The authors found that the tumors in the under 50 group are molecularly distinct from tumors found in older patients, while at the same time they can even vary among subsets of the younger adults. More specifically, they observed unique signaling aberrations among younger patients, aged 18 to 29 years, versus other early-onset patients, aged 30 to 49 years, and among early-onset patients with predisposing conditions, including inflammatory bowel disease or a hereditary syndrome, versus patients without these conditions. In the younger patients, the occurrence of BRAF V600 mutations was significantly lower, and the number of KRAS mutations was much but not significantly lower, and the occurrence of combined MAPK pathway mutations were the lowest in 18–29 year old patients compared with other age groups.

### 4.3. Impact on Guidelines and Health Care Policy

Current guidelines in Europe recommend CRC screening starting from the age of 50. In 2018, the American Cancer Society recommended to start screening at the age of 45 [12]. This recommendation was based on the burden of disease, the increasing incidence among younger subjects, the results of modelling, and the assumption that screening of the age group 45–49 years will have a preventive effect similar to screening those 50 years and above. Other global and European studies reporting an increasing incidence of CRC in young adults over the last 25 years also support starting screening at age 45 in Europe [11,23].

A high priority of the Cancer Registry of Crete is to direct future research to this young group with CRC and explore if these cases are in part due to heredity cancer syndromes or are mainly sporadic. Efforts should also be undertaken to make clinicians aware that the CRC incidence in young adults is rising quite rapidly. The role of primary health care services that are currently under reform seems to be important, and health care policy makers could consider including screening for CRC to be initiated at the time of contract with the family physician. Both statements are in agreement with the recommendations that appeared in the Gut paper2, underscoring the role of primary care physicians in the evaluation of family history and symptoms of young individuals. This paper also highlights the need for research on early-life exposures in relation to colorectal carcinogenesis [23].

### 4.4. Strengths and Limitations

The findings of this study should be discussed carefully and in light of certain limitations. Current data and results concern the region of Crete exclusively, in which we had high coverage and quality indicators, and not the entire country of Greece. Another limitation of the present study, like any other population-based study of impact data, is that there is a possibility that some cases may have been “lost” during sample collection or have been incorrectly recorded. This is a result of possible errors in completing death certificates. However, an attempt was made to manage this limitation by calculating weighted incidence rates rather than studying absolute numbers. In addition, similar errors were reduced through the methodological framework adopted by the registry, which follows international standards of recording, classification, and analysis. With these standard procedures, it has managed to reduce the probability of error to 3.8%, a fact that is not expected to significantly affect the statistical findings or the drawing of conclusions about the trends under study.

Contrary to the above limitations, this work is the first comprehensive study on the incidence of colorectal cancer, highlighting the trends of recent years at the local level and the risk factors. In fact, the large sample size and the long coverage period (1992–2021) are key strengths of the study. At the level of the health district, the coverage of the new cases has been done with quite high accuracy as mentioned in the methodology. The selected data analysis methodology used spatial and temporal models, enabling high internal reliability. Future studies of our team will focus on the exploration of the determinants of colorectal cancer in younger adults and the associated factors of the observed increases among them versus older adults. Additionally, we will attempt to open the discussion about earlier screening in Greece and assess beliefs and perceptions of young adults, physicians, and policy makers.

## 5. Conclusions

This research has stressed that the incidence of CRC in Crete is rising among subjects 20–49 years similar to the published evidence from other European regions. It invites the healthcare authorities of Crete to monitor this trend and undertake actions towards increasing the awareness of both healthcare providers and the public, and to contribute to the current dialogue on the need to update and reform the European screening programs.

## Figures and Tables

**Figure 1 cancers-14-04799-f001:**
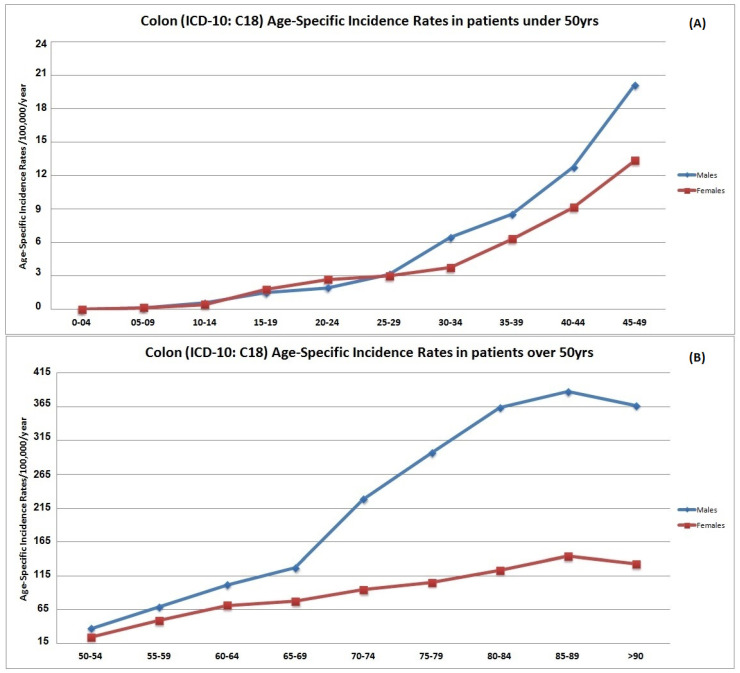
Age-specific incidence rates (ASpIR) per 100,000 population per year in colon cancer patients under (**A**) and over (**B**) 50 years.

**Figure 2 cancers-14-04799-f002:**
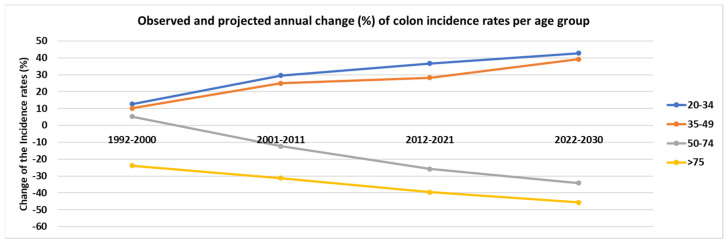
Observed and projected annual change (%) of colon cancer incidence rates per age group.

**Figure 3 cancers-14-04799-f003:**
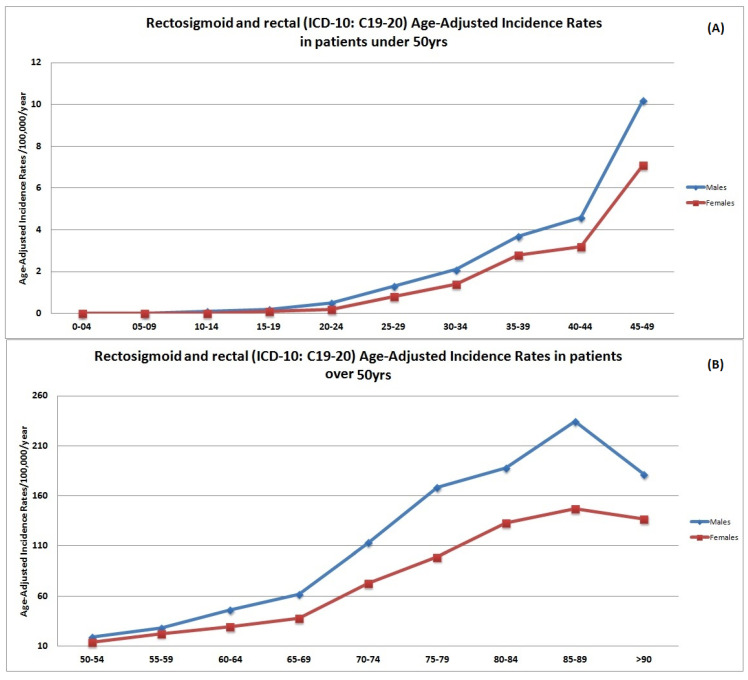
Age-specific incidence rates (ASpIR) per 100,000 population per year in rectal cancer patients under (**A**) and over (**B**) 50 years.

**Figure 4 cancers-14-04799-f004:**
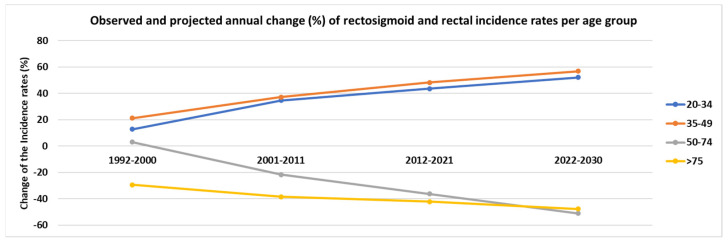
Observed and projected annual change (%) of rectal cancer incidence rates per age group.

**Table 1 cancers-14-04799-t001:** Characteristics of the study sample and annual percentage change of the ASIR (age-standardized incidence rates).

Characteristics	N (%)	Annual Percentage Change (95% CI)
**Colon cases**	2895 (75.1)	1.5 (1.3–1.7) ^a^
**Rectosigmoid junction and rectal cases**	962 (24.9)	0.9 (0.0–1.6)
**Gender**		
*Males*	2141 (55.5)	0.7 (0.5–0.9) ^a^
*Females*	1716 (44.5)	1.8 (1.4–2.2) ^a^
**Age**		
*20–34*	42 (1.1)	1.8 (1.2–2.6) ^a^
*35–49*	116 (3.0)	1.6 (1.3–1.9) ^a^
*50–74*	1137 (29.5)	−1.1 (−1.4–−0.5) ^a^
*>75*	2562 (66.4)	−1.4 (−1.3–−1.0) ^a^
**Stage at diagnosis**		
*Stage I*	497 (12.9)	0.2 (0.0–0.4) ^a^
*Stage II*	927 (24.0)	2.2 (1.9–2.2) ^a^
*Stage III*	1269 (32.9)	1.3 (1.1–1.5) ^a^
*Stage IV*	817 (21.2)	−2.2 (−2.3–−2.1) ^a^
*Stage unknown*	347 (9.0)	-
**Place of residence**		
*County of Heraklion*	1193 (30.9)	0.7 (0.6–1.1) ^a^
*County of Lasithi*	1121 (29.1)	1.9 (1.7–2.3) ^a^
*County of Rethymnon*	697 (18.1)	1.4 (1.2–1.6) ^a^
*County of Chania*	846 (21.9)	0.8 (0.5–1.2)

^a^*p*-value < 0.05.

**Table 2 cancers-14-04799-t002:** Common risk factors among younger adults (under 50 years old) versus older adults (over 50 years old) and statistically significant changes among the three last decades (1992–2000, 2001–2011, 2012–2021).

Risk Factors	Time Period RR ^a^ (95% CI)			*p*-Value for Interaction ^b^
	1992–2000	2001–2011	2012–2021	
**Body Mass Index (kg/m^2^) ***	1.2 (1.1–1.3)	1.8 (1.4–2.1)	2.5 (2.2–2.7)	0.04
**BSA (m^2^) ***	1.4 (1.1–1.8)	2.3 (2.1–2.6)	3.4 (3.3–3.5)	<0.001
**Smoking status**				
*Never smokers*	1	1	1	
*Ever smokers (ex and current)*	1.7 (1.4–2.1)	1.8 (1.4–2.2)	2.1 (2.0–2.2)	0.58
**Packyears**	1.8 (1.5–2.2)	2.5 (2.3–2.7)	3.1 (3.0–3.2)	0.01
**Alcohol consumption**				0.02
*Never consumers*	1	1	1	
*Ever consumers (ex and current)*	1.9 (1.7–2.2)	2.6 (2.4–2.8)	3.3 (3.0–3.4)	
**Number of glasses per week**	2.1 (1.9–2.3)	2.9 (2.7–3.1)	4.1 (3.9–4.3)	<0.001
**Occupation**				0.04
*Other occupations*	1	1	1	
*Farmers*	1.5 (1.4–1.6)	1.8 (1.7–2.0)	(2.1–2.5)	

^a^ Adjusted Relative Risk to gender, place of residence, comorbidities of the gastroenterological system, and cancer family history; ^b^
*p*-value for within stratum test by Mantel–Haenszel procedure; Yate’s corrected (Ho: p1 = p2). * While body mass index (BMI) is a measure of a person’s body fat mass, body surface area (BSA) measures the total surface area of a person’s body and is frequently used in order to calculate drug dosage and the amount of fluids to be administered. The average adult BSA is 1.7 m^2^ (1.9 m^2^ for adult males and 1.6 m^2^ for adult females).

**Table 3 cancers-14-04799-t003:** Statistical differences on survival, stage at diagnosis, metastatic cancer, and therapeutic approach between younger and older adults.

Selected Parameters	Younger Adults	Older Adults	*p*-Value
**5-year Net survival** (**%**)			
*Period 1992–2000*	67.0	58.4	0.03
*Period 2001–2011*	64.1	60.5	
*Period 2012–2021*	61.2	61.9	
**Stage at diagnosis** (Stage III & IV)			
*Period 1992–2000*	19.8	27.4	0.03
*Period 2001–2011*	27.3	26.1	
*Period 2012–2021*	29.1	25.7	
**Metastatic cancer (yes)**			
*Period 1992–2000*	23.1	25.8	0.72
*Period 2001–2011*	24.7	25.7	
*Period 2012–2021*	25.2	25.1	
**Treatment (Surgery)**			
*Period 1992–2000*	63.1	63.2	0.81
*Period 2001–2011*	64.7	63.1	
*Period 2012–2021*	64.7	63.8	
**Treatment (Chemotherapy)**			
*Period 1992–2000*	32.7	42.5	0.04
*Period 2001–2011*	38.9	38.0	
*Period 2012–2021*	43.2	32.5	
**Treatment (Radiotherapy)**			
*Period 1992–2000*	18.6	20.9	0.52
*Period 2001–2011*	21.5	21.6	
*Period 2012–2021*	22.0	22.9	

## Data Availability

Data could only be available for research purposes, upon request from the corresponding author.

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
