# Peer review of "Significant Rise of Colorectal Cancer Incidence in Younger Adults and Strong Determinants: 30 Years Longitudinal Differences between under and over 50s"

_cancers, 2022, doi:10.3390/cancers14194799_

Round 1

Reviewer 1 Report

This is a well-written and well-structured paper, easy to read and to understand. The paper discusses the differences between colorectal cancer incidences in younger adult vs elders the last 3 decades. The paper’s originality is based on the presentation of outcomes from a large sample of the Greek population and on the calculation of specific indicators (ex. AsIR/100,000/year). No novelty presented on the methods or statistical analyses.

Some suggestions:

1) It is more than welcome to give a short presentation of the indicators that you used in the Introduction section.

2) Figure 1 must be resubmitted in a higher resolution.

3) Future work may be added after the study’s limitations.

Author Response

This is a well-written and well-structured paper, easy to read and to understand. The paper discusses the differences between colorectal cancer incidences in younger adult vs elders the last 3 decades. The paper’s originality is based on the presentation of outcomes from a large sample of the Greek population and on the calculation of specific indicators (ex. AsIR/100,000/year). No novelty presented on the methods or statistical analyses.

Re: We would like to thank you for reviewing the paper and we have provided detailed responses below.

Some suggestions:

  • It is more than welcome to give a short presentation of the indicators that you used in the Introduction section.

Re: Thank you for the comment. We have added a relevant sentence in the introduction and further explain the indicators used in the methods section.

  • Figure 1 must be resubmitted in a higher resolution.

Re: We have enhanced the resolution. Thank you.

3) Future work may be added after the study’s limitations.

Re: Future studies of our team will focus on the exploration of the determinants of colorectal cancer in younger adults and the associated factors of the observed increases among them versus older adults. Additionally, we’ll attempt to open the discussion about earlier screening in Greece and assess beliefs and perceptions of young adults, physicians and policy makers.

Reviewer 2 Report

#1 p2-3 “clinical risk factors [14]…. Registries (ENCR) [25].” >> or Registries (ENCR) [15] ?

#2. section 2.2 “For the needs of this research, the study population was defined as the total number of permanent residents (total population based on the 2011 census: 623,065). Data on malignant neoplasms of colon (ICD-10: C18) and rectosigmoid junction and rectum (ICD-10: C19-20) were obtained from the database of the regional Cancer Registry of Crete for the period 1992-2021” >> Was 623,065 used as the population size irrespective of the incident year[a1] ? Please clarify and justify.

#3. section 2.4 “validity and completeness according to the European guidelines [26]” >> or the European guidelines [16]

#4. section 2.5 “The ASIR annual percentage change (APC) was estimated per year, while projection of the expected changes was attempted using a Bayesian model” >> Please provide reference[s] and more details.

#5. section 2.6 “will be published.3.” >> or “will be published.” ?

#6. table 1: the APC were 0.9 for rectum and 1.5 for colon, but <=0.9 for all counties, were them consistent?

#7. page 5 “ASpIR” >> What is ASpIR? Please specify the details in the method. (or AsIR in section 2.5 in fact?)

#8. discussion “occupational exposures [29” but re-29 = Familial colon cancer syndromes: An update of a rapidly evolving field., so please confirm.

#9. discussion “dietary habits [31], including consumption of more alcohol and processed meat [33].” >> so where was ref-32 cited in the text?

Author Response

#1 p2-3 “clinical risk factors [14]…. Registries (ENCR) [25].” >> or Registries (ENCR) [15] ?

 Re: Thank you we corrected it.

#2. section 2.2 “For the needs of this research, the study population was defined as the total number of permanent residents (total population based on the 2011 census: 623,065). Data on malignant neoplasms of colon (ICD-10: C18) and rectosigmoid junction and rectum (ICD-10: C19-20) were obtained from the database of the regional Cancer Registry of Crete for the period 1992-2021” >> Was 623,065 used as the population size irrespective of the incident year[a1] ? Please clarify and justify.

 Re: Yes, this was the population size used for standardization. We have now clarified it in the text.

#3. section 2.4 “validity and completeness according to the European guidelines [26]” >> or the European guidelines [16]

Re: Thank you we corrected it.

#4. section 2.5 “The ASIR annual percentage change (APC) was estimated per year, while projection of the expected changes was attempted using a Bayesian model” >> Please provide reference[s] and more details.

  Re: We have added a reference  

#5. section 2.6 “will be published.3.” >> or “will be published.” ?

 Re: We’ve omitted the “3”. Thank you.

#6. table 1: the APC were 0.9 for rectum and 1.5 for colon, but <=0.9 for all counties, were them consistent?

  Re: Many thanks for noticing. It was our mistake, a mis-typing of the results at that point. We corrected them and checked all results once again.

#7. page 5 “ASpIR” >> What is ASpIR? Please specify the details in the method. (or AsIR in section 2.5 in fact?)

  Re: These are two different rates that are explained in the methods. ASIR stands for Age-Standardized Incidence Rates and ASpIR stands for Age-Specific Incidence Rates. We’ve now added their explanation also in the results in the title of Table 1 and Figure 1 and 3.  

#8. discussion “occupational exposures [29” but re-29 = Familial colon cancer syndromes: An update of a rapidly evolving field., so please confirm.

 Re: We replaced this reference with Duijster, J., Mughini-Gras, L., Neefjes, J., & Franz, E. (2021). Occupational exposure and risk of colon cancer: a nationwide registry study with emphasis on occupational exposure to zoonotic gastrointestinal pathogens. BMJ open, 11(8), e050611.

#9. discussion “dietary habits [31], including consumption of more alcohol and processed meat [33].” >> so where was ref-32 cited in the text?

Re: We have added this reference in the text. No no.32 is 33 since we’ve changed the order of some references.

Round 2

Reviewer 2 Report

# reference-20 . Available from: http://www.who.int/healthinfo/paper31.pdf >> but the above hyperlink did not work and showed “This page cannot be found. The page or file you are trying to access cannot be found. This is because the web address is incorrect or the file has been moved or deleted.. In 2020, we migrated our web content to a new system so some older content may no longer be available online or at the same place.” >> was it actually https://cdn.who.int/media/docs/default-source/gho-documents/global-health-estimates/gpe_discussion_paper_series_paper31_2001_age_standardization_rates.pdf ?

Author Response

Dear reviewer, thank you for noticing. We have revised the link and the reference. 
